# Anti-Metastatic Effect of Pyruvate Dehydrogenase Kinase 4 Inhibition in Bladder Cancer via the ERK, SRC, and JNK Pathways

**DOI:** 10.3390/ijms232113240

**Published:** 2022-10-31

**Authors:** Eun Hye Lee, Jae-Wook Chung, Eunji Sung, Bo Hyun Yoon, Minji Jeon, Song Park, So Young Chun, Jun Nyung Lee, Bum Soo Kim, Hyun Tae Kim, Tae Hwan Kim, Seock Hwan Choi, Eun Sang Yoo, Tae Gyun Kwon, Ho Won Kang, Wun-Jae Kim, Seok Joong Yun, Sangkyu Lee, Yun-Sok Ha

**Affiliations:** 1Joint Institute of Regenerative Medicine, Kyungpook National University, Daegu 41566, Korea; 2Department of Urology, School of Medicine, Kyungpook National University, Daegu 41405, Korea; 3BK21 FOUR Community-Based Intelligent Novel Drug Discovery Education Unit, College of Pharmacy, Research Institute of Pharmaceutical Sciences, Kyungpook National University, Daegu 41566, Korea; 4Division of Biotechnology, Daegu Gyeongbuk Institute of Science and Technology, Daegu 42988, Korea; 5BioMedical Research Institute, Kyungpook National University Hospital, Daegu 41944, Korea; 6Department of Urology, College of Medicine, Chungbuk National University, Cheongju 28644, Korea; 7Department of Urology, Chungbuk National University Hospital, Cheongju 28644, Korea; 8Institute of Urotech, Cheongju 28120, Korea

**Keywords:** bladder cancer, metastasis, invasion, muscle-invasive bladder cancer (MIBC), pyruvate dehydrogenase kinase 4 (PDK4)

## Abstract

Bladder cancer is a common global cancer with a high percentage of metastases and high mortality rate. Thus, it is necessary to identify new biomarkers that can be helpful in diagnosis. Pyruvate dehydrogenase kinase 4 (PDK4) belongs to the PDK family and plays an important role in glucose utilization in living organisms. In the present study, we evaluated the role of PDK4 in bladder cancer and its related protein changes. First, we observed elevated PDK4 expression in high-grade bladder cancers. To screen for changes in PDK4-related proteins in bladder cancer, we performed a comparative proteomic analysis using PDK4 knockdown cells. In bladder cancer cell lines, PDK4 silencing resulted in a lower rate of cell migration and invasion. In addition, a PDK4 knockdown xenograft model showed reduced bladder cancer growth in nude mice. Based on our results, PDK4 plays a critical role in the metastasis and growth of bladder cancer cells through changes in ERK, SRC, and JNK.

## 1. Introduction

Bladder cancer, also known as urinary bladder cancer, is the 10th most common cancer worldwide [1] and one of the most frequent cancers of the urinary tract [2]. It accounts for approximately 500,000 new cases and 200,000 deaths annually [3]. Approximately 75% of newly diagnosed cases belong to non-muscle-invasive bladder cancers, whereas the remaining 25% are muscle-invasive bladder cancers (MIBC) or metastatic diseases [4]. Male sex, aging, Caucasian race, family history, pelvic radiation, cigarette smoking, family history, and chronic bladder infection/irritation are established risk factors for bladder cancer [5]. Cisplatin-based combination therapies are the standardized first-line treatments for bladder cancer; however, only 50% of the cases respond to the therapy [6]. For metastatic bladder cancer, the treatment options have been improved owing to second-line immunotherapies; however, only 20–30% of cases show partial or complete response to checkpoint immunotherapies [7].

For the maintenance of homeostasis, viability, growth, metabolism, and proper function, cells require chemical energy in the form of adenosine triphosphate (ATP). Glycolysis, the foundation of cellular metabolism, is the process of converting glucose to energy in the cytoplasm under anaerobic conditions [8]. During aerobic glycolysis, pyruvate enters the mitochondrial tricarboxylic acid (TCA) cycle and produces ATP [9]. Although normal cells produce energy through pyruvate oxidation, most proliferating cancer cells metabolize glucose through aerobic glycolysis; this metabolic reprogramming is called the Warburg effect or aerobic glycolysis [10]. Cancer cells maintain higher rates of glycolysis than normal cells [9]. Pyruvate dehydrogenase kinase 4 (PDK4) belongs to the PDK family (PDK1–4) that is located in the mitochondrial matrix [11] and controls the pyruvate dehydrogenase (PDH) complex [10]. PDK2 and PDK4 are the most widely expressed isoforms of PDKs [12]. PDK4 restricts the entry of pyruvate into the TCA cycle by suppressing PDH activity [11]; its gene is located on chromosome 7q21.3, with 11 exons and encoding a protein of 411 amino acids [13]. PDK4 is reported to be involved in diverse cancers, such as colon, lung, and breast cancers [9,11,14]. PDK4 inhibition stimulates the mitochondrial metabolism of pyruvate in cancer cells, which may suppress cancer cell proliferation [15].

Based on advanced studies, PDK4 is reported to be related to human colon, breast, ovarian, and gastric cancers [9,11,13,16]. In human colon cancer cells, inhibition of PDK4 results in decreased migration, invasion, and resistance to apoptosis [9]. In addition, patients with breast cancer and high PDK4 expression have poor prognosis and survival rates [11]. In ovarian cancer cells, the overexpression of PDK4 promotes cell proliferation and invasion. PDK4 is also associated with chemoresistance [13]. PDK4 is highly expressed in gastric cancer cells, and PDK4 is significantly correlated with tumor-infiltrating immune cells [16]. However, the effect of PDK4 on bladder cancer has not been studied.

Proteomic analysis is a great experimental tool for studying protein changes in cells, tissues, or organisms [17]. Recently, proteomic analysis has been widely used to enable the discovery of potential biomarkers and identification of tumor-related protein expression [18,19,20]. Because comparative phosphoproteome analysis can comprehensively quantify altered protein phosphorylation, it is effective to evaluate the intracellular effect of PDK4 as a kinase. In this study, we demonstrated that PDK4 is involved in the proliferation, migration, and invasion of bladder cancer cells. Inhibition of PDK4 suppressed migration and invasion in vitro and tumor growth in vivo through the extracellular signal-regulated kinase (ERK), SRC, and JNK pathways. Furthermore, we evaluated the overexpression of PDK4 in human bladder cancer samples and its expression pattern in different cancer stages.

## 2. Results

### 2.1. Effects of PDK4 Inhibition on Bladder Cancer Cell Lines

To verify the relationship between PDK4 and tumorigenesis in bladder cancer, we suppressed PDK4 expression as a first step. We confirmed the suppressed mRNA (Figure 1A) and protein (Figure 1B) expression of PDK4 in two bladder cancer cell lines, T24 and J82, by siRNA transfection using qPCR and Western blot analyses. After validating the PDK4 knockdown, we performed migration and invasion assays with the T24 and J82 cell lines. In the PDK4 siRNA-treated groups of both cell lines, we observed a reduced number of cells in the migration (Figure 2A) and invasion (Figure 2B) assays. In the measurement of migrating and invading cell detection, PDK4 siRNA-treated groups showed a significantly suppressed number of cells in migration and invasion assays.

### 2.2. Mechanism of PDK4 in Bladder Cancer Cell Lines

To determine the underlying mechanism of PDK4 in bladder cancer, we evaluated the expression of migration- and invasion-related proteins, p-ERK p-SRC, and p-JNK (Figure 3). p-ERK at highly increased concentration was observed in both cell lines. In the T24 cell lines, p-SRC and p-JNK were decreased, and in the J82 cell lines, only p-JNK showed decreased expression.

### 2.3. Anti-Tumorigenic Effect of PDK4 Inhibition in a Xenograft Model

To confirm the antitumor effect of PDK4 inhibition in vivo, we performed xenograft experiments with immune-suppressed mice. Seven days after injection, the differences in tumor size between the wild-type and PDK4-inhibited cells were significant (Figure 4A). The tumor sizes of the control and PDK4 knockdown groups were 1300.573 mm^3^ and 772.085 mm^3^, respectively, at the end of the experiment (Figure 4B). After collecting the tumor samples, we analyzed the protein expression of PDK4, p-ERK, p-SRC, and p-JNK by IHC staining. As expected from the in vitro results, the PDK4 knockdown group showed decreased levels of PDK4, p-SRC, and p-JNK, and elevated levels of p-ERK (Figure 4C).

### 2.4. Expression of PDK4 in Human Bladder Tumor Biospecimens

We further examined PDK4 expression in different stages of human bladder cancer to evaluate the association between PDK4 and tumor prognosis. In quantitative analysis, PDK4 mRNA expression in T3 bladder cancer tissues was twofold higher than that in normal tissues (Figure 5A). Moreover, PDK4 was upregulated in bladder cancer tissues and its expression was further increased in advanced tumor tissues. IHC results showed that T3 bladder cancer tissues had higher levels of PDK4 expression than normal tissues (Figure 5B).

### 2.5. Proteomic Screening of Phosphorylation in Bladder Cancer Cells

To visualize the differential protein patterns, whole protein from bladder cancer cell lines was analyzed using sodium dodecyl sulfate-polyacrylamide gel electrophoresis (SDS-PAGE) (Appendix A). The protein expression levels in the cell lysates of the T24 and J82 groups were similar. The presence of phosphorylated proteins in each group was confirmed by Western blotting using anti-Ser/Thr/Tyr-phosphorylation antibodies. The results show that the phosphorylation levels in T24 and PDK4 knockdown (KD) T24 cells were comparable. In contrast, phosphorylation was clearly reduced in PDK4 KD J82 cells compared with that in PDK J82 control cells. Based on these results, we conducted global phosphoproteome analysis in J82 cells, and a schematic diagram of the experimental design is shown in Figure 6A. We performed LC-MS/MS analysis combined with ^18^O labeling and TiO_2_ enrichment technology to compare the relative quantitation of phosphoproteomes between the J82 control and PDK4 KD J82 cells. Sample preparation was conducted using technically duplicated experiments (R2 = 0.913) (Appendix A). Our data show that the phosphorylated peptides were ^18^O labeled with high reproducibility, with an efficiency of 98.4%.

In total, 3743 phosphosites, corresponding to 1670 phosphoproteins, were identified. We overlapped the specifically expressed phosphosites between the J82 and PDK4 KD J82 groups (Figure 6B). As shown in the Venn diagrams, 202 phosphosites were present in PDK4 KD J82, 30 in J82, and 1362 phosphosites in both. Among them, 30 were detected only in J82 control cells and could be potential substrates for PDK4 because they were not detected in PDK4 KD J82 cells. A total of 209 differentially regulated normalized phosphopeptides were identified using the volcano plot. In PDK4 KD J82 cells, the levels of 130 phosphopeptides were significantly increased, and those of 79 phospho-peptides were decreased compared to J82 control cells (Figure 6C and Appendix A). The levels of phosphopeptides with a quantitative ratio greater than 1.5 were considered upregulated, and those smaller than 0.66 were deemed downregulated.

Decreased phosphorylation in PDK4 KD J82 cells or detected only in J82 control cells can be considered candidate PDK4 substrates. To acquire an in-depth understanding of the biological functions of potential PDK4 substrates, we used DAVID web-based software and conducted an enrichment analysis of different categories, including gene ontology (GO) and KEGG pathways (Appendix A). KEGG pathway enrichment analysis revealed that ribosome biogenesis in eukaryotes, adherens junctions, proteoglycans in cancer, bladder cancer, and gap junctions were significantly enhanced. The bladder cancer category included proteins such as MAPK1 (P28482), EGFR (E9PFD7), SRC (P12931), and ERBB2 (B4DTR1). In addition, to further investigate kinase substrate relationships, a comprehensive KSPN for the differentially regulated phosphopeptides was constructed using iGPS 1.0. The results show that ANXA2 and MPZL1 were detected as substrates, FYN kinase was detected, and both SRC regulatory kinase and SRC were detected.

## 3. Discussion

MIBC (T2-T4) has a poor prognosis and requires complex therapy with radical surgery, radiotherapy, and chemotherapy. After the 1960s, radical cystectomy was set as a standardized therapy for MIBC patients without signs of metastasis [21]. Cisplatin-based neoadjuvant chemotherapy has been used for MIBC treatment because it showed increased overall survival, from 30% to 36% after 10 years [22]. In the case of a T2 grade MIBC, which has the possibility of complete tumor removal owing to its small size, chemoradiotherapy might be an alternative [23]. However, almost half of these patients show metastases and die within 5 years [24].

Recently, neoadjuvant chemotherapy has been reported to improve the overall survival rate of patients with MIBC; however, it remains an important and urgent issue to select biomarkers that predict resistance against cisplatin-based treatment and to understand the underlying mechanisms. Existing clinical variables, such as tumor pathological stage, histology class, and local lymph node stage, have been reported to be insufficient to predict prognosis. It would be useful for patients and therapists to choose bladder conservation treatments, such as mild urethral bladder tumor resection or partial bladder resection with chemoradiotherapy and have biomarkers that can predict treatment responses [25].

The incompleteness of the currently used pathological prognostic factors requires a clinical application of molecular biology-based biomarkers [26]. However, researched biomarkers have not yet been used in clinical practice. Currently, it is necessary to improve the treatment of MIBC and apply customized treatment; however, there are no biomarkers or panels that can be applied to clinical trials to predict the prognosis of this disease or to select an appropriate treatment [27].

Various bladder cancer treatments are available; however, the long-term prognosis of patients with metastatic bladder cancer remains poor. Angiogenesis, metastasis, and invasion of cancer cells are key risk factors for poor survival of patients with bladder cancer [3]. Thus, the present study evaluated the PDk4 protein as a biomarker in bladder cancer by investigating its function in this disease. To evaluate the role of PDK4 in bladder cancer, we suppressed PDK4 expression, and downregulated PDK4 reduced the migratory capacity and invasiveness of bladder cancer cells by modulating p-ERK, p-SRC, and p-JNK. In addition, we examined the relationship between PDK4 and tumor growth in vivo, as well as in human bladder cancer specimens.

PDK4 is identified as a chemoresistance-associated gene in the gene expression omnibus (GEO) database [13]. PDK4 phosphorylates and inhibits pyruvate oxidation, thereby preventing the oxidation of glucose-derived carbon. This leads to a metabolic switch from glucose to fatty acid metabolism. Cancer cells, which grow rapidly, require carbon aside from ATP for biosynthesis [28]. Thus, we evaluated the expression of PDK4 in human bladder cancer tissues and performed proteomic analysis to identify PDK4-related proteins in bladder cancer. According to proteomic analysis of PDK4 knockdown cells, SRC appears to be a PDK4-related protein. In living organisms, ERK, SRC, and JNK are master proteins that modulate cellular physiological processes such as inflammatory responses, cell proliferation, differentiation, cell survival, and apoptosis [29,30,31]. Activated ERK (p-ERK) exerts anti-tumorigenic effects in multiple cancers [32,33,34]. Activated SRC (p-SRC) is elevated in human ovarian, colorectal, breast, and lung cancers [35,36,37,38]. Previous studies have shown that activated JNK (p-JNK) promotes progression in colon, breast, and prostate cancers [39,40,41,42]. JNK exerts an anti-tumorigenic effect by generating ROS, causing cell cycle arrest, cell apoptosis, and autophagy [42,43,44,45]. These three proteins are well-known cancer-related proteins [46,47,48].

Metastatic bladder cancer has a low response rate to chemotherapy and immune checkpoint inhibitors; thus, we designed this study to discover new biomarkers and candidates for targeted therapy. Our results show that PDK4 was overexpressed in human bladder cancer tissues. We then suppressed PDK4 expression in bladder cancer cell lines, which resulted in suppressed metastasis as seen in the reduced number of migrating and invading cancer cells. Moreover, PDK4 knockdown altered the expression of p-ERK, p-SRC, and p-JNK, which are related to migration and invasion. To clarify the effect of PDK4 in vivo, we used a xenograft model that showed reduced tumor size in PDK4 knockdown cells. As in human bladder cancer specimens, PDK4 expression was correlated with the T category in bladder cancer patients. We also performed proteomic analysis of J82 cell lines (control and PDK4 KD) and observed significant expression changes in p-SRC. Taken together, these results demonstrate the functional and physical relationship between PDK4 and tumor growth and metastasis in bladder cancer. Further studies are required to explore the underlying mechanisms and related proteins to understand the role of PDK4 in multiple types of cancers.

## 4. Materials and Methods

### 4.1. Reagents

Anti-phospho-SRC, anti-SRC, anti-phospho-ERK, anti-ERK, anti-phospho-JNK, and anti-JNK antibodies were obtained from Cell Signaling Technology (Danvers, MA, USA). Anti-PDK4 antibody was purchased from Novus Biologicals (Littleton, CO, USA). Dimethyl sulfoxide (DMSO), crystal violet, a cell migration kit, and a cell invasion kit were purchased from Sigma-Aldrich (St. Louis, MO, USA).

### 4.2. Cell Lines and Culture

The bladder cancer cell lines T24 and J82 were purchased from ATCC. Cells were maintained in DMEM/high media (T24) and RPMI 1640 (J82) with 10% FBS, 50 units/mL penicillin, and 50 µg/mL streptomycin (Gibco, Waltham, MA, USA) at 37 °C in a humidified 5% CO_2_ atmosphere.

### 4.3. siRNA Transfection

ON-TARGETplus human PDK4 siRNA SMART pool, non-targeting pool, and siRNA transfection reagents were purchased from Dharmacon (Lafayette, CO, USA). The four sequences of PDK4 were 5′-GAGCAUUUCUCGCGCUACA-3′, 5′-CGACAAGAAUUGCCUGUGA-3′, 5′-CAACGCCUGUGAUGGAUAA-3′, and 5′-GACCGCCUCUUUAGUUAUA-3′. Transfection reagent, non-targeting siRNA, or PDK4 siRNA were used to treat T24 and J82 cells. After 12 h, culture medium was changed to growth medium and incubated for 36 h at 37 °C in a humidified 5% CO_2_ atmosphere.

### 4.4. shRNA Transfection

To produce a PDK4 knockdown stable cell line, pLKO.1-scramble and pLKO.1-shPDK4 vectors were obtained from Sigma-Aldrich. J82 cells were cultured in 10% FBS, 50 units/mL penicillin, and 50 μg/mL streptomycin (Hyclone Laboratories, Logan, UT, USA) containing MEM-alpha media. J82 cells were seeded in six-well plates (5 × 10^6^) and incubated with packaging plasmids (pMDLg/pRRE, pMD2.G, and pRSV-Rev) and Fugene HD reagents with pLKO.1-scramble and pLKO.1-shPDK4 vector for 24 h. After incubation, the medium was changed, and the cells were harvested after 48 h. To collect the lentivirus, the harvested cell culture supernatant was filtered through a 0.45 um pore size filter. The collected lentivirus was added to J82 cell culture media with 8 μg/mL of polybrene (Sigma-Aldrich), and cells were incubated for 48 h. Puromycin at 1.5 µg/mL was used to distinguish uninfected cells.

### 4.5. Protein Extraction

The cell pellets were washed with phosphate buffer saline (PBS; 138 mM NaCl, 2.7 mM KCl, 1.47 mM KH_2_PO_4_, and 8.0 mM Na_2_HPO_4_ pH 7.3) and lysed in RIPA buffer (Thermo Fisher Scientific, Waltham, MA, USA) containing a protease inhibitor cocktail (Thermo Fisher Scientific) and a phosphatase inhibitor (Thermo Fisher Scientific). The cell lysate was sonicated for 1 min (cycles of 5 s on and 5 s off) on ice and centrifuged at 16,000× *g* for 7 min to remove the cell debris. Supernatants were collected in low-binding tubes. The protein concentration was determined using the BCA protein assay (Thermo Fisher Scientific). The extracted protein was kept at −80 °C until use.

### 4.6. Immunoblot Analysis

Treated cells were incubated with RIPA buffer at 4 °C and centrifuged at 15,000× *g* for 30 min and 4 °C. Lysates were collected and proteins were quantified using the Pierce BCA Protein Assay Kit (Thermo). Protein samples were loaded into SDS-PAGE and transferred to PVDF membranes. Primary antibodies (1:1000) were incubated overnight at 4 °C and secondary antibodies (1:5000) were incubated for 2 h at room temperature. The membrane was developed using ECL reagent (Advensta, Menlo Park, CA, USA) and images were captured with a chemi-doc image analyzer (iBright 1500, Thermo Fisher Scientific).

### 4.7. In-Solution Digestion and Enzymatic ^18^O Labeling

For each sample, 3 mg of protein was reduced in 5 mM dithiothreitol (DTT) at 56 °C for 30 min and alkylated in 15 mM iodoacetamide (IAA) at room temperature for 30 min in the dark. Pure proteins were obtained via trichloroacetic acid/acetone precipitation. For trypsin digestion, protein pellets were resuspended in 50 mM ammonium bicarbonate (ABC) buffer and proteolytically cleaved with 20 μg of sequencing-grade trypsin (Promega Corporation, Madison, WI, USA) at a ratio of 50:1 (protein:enzyme, *w*/*w*) at 37 °C overnight. The digested peptides were dried using a speed-vac and immediately re-dissolved in 50 µL of 50 mM calcium chloride, followed by the addition of trypsin (Promega Corporation, Madison, WI, USA) at a ratio of 50:1 (protein:enzyme, *w*/*w*). For enzymatic labeling, peptides were differentially labeled with either ^16^O or ^18^O water (97%, Cambridge Isotope Laboratories, MA, USA) containing 20% acetonitrile and incubated for 24 h at 37 °C. Labeling was stopped using 1% TFA. Peptide concentration was measured using a quantitative colorimetric peptide assay kit (Thermo Fisher Scientific). For comparison, two labeled samples were mixed in equal amounts and dried.

### 4.8. Phospho-Peptide Enrichment and Desalting

The mixture of labeled peptides was enriched using TiO_2_ phosphopeptide enrichment and a clean-up kit (Thermo Fisher Scientific) according to the manufacturer’s instructions. Additionally, the cleaned-up peptide mixture was desalted using SDB and GC tips (GC Science, Tokyo, Japan) according to the manufacturer’s instructions. The desalted peptide mixture was then dried using a speed-vac and stored at −80 °C before analysis.

### 4.9. LC-MS/MS Analysis

Peptide samples were analyzed on an UltiMate 3000 RSLC Nano LC system (Thermo Fisher Scientific), coupled with a LTQ-Orbitrap mass spectrometer (Thermo Fisher Scientific). The samples were dissolved in solvent A (0.1% formic acid) and loaded onto a trap column (PepMap 100 trap column, 75 µm × 2 cm, C18, 3 um, 100 Å, Thermo) and separated at a flow rate of 300 μL/min with a 180 min gradient (buffers A and B: 5% Solvent B for 7 min, 30% Solvent B for 137 min, 80% Solvent B for 140–160 min, and 5% Solvent B for 180 min). The MS data were obtained in collision-induced dissociation (CID) mode. The electrospray voltage was set at 2.0 kV. For the MS1 full scan, ions with *m*/*z* ranging from 300 to 1800 were acquired at a high resolution of 60,000. The automatic gain control (AGC) was set as 1.0 × 10^6^. The maximum IT was 100 ms for the full scan, and 50 ms for MS2. Data are available via ProteomeXchange with the identifier PXD035005 [49].

### 4.10. Database Search and Bioinformatics Analysis

The MS raw files were processed using MaxQuant (version 1.5.1.0) software against he UniProt homo sapiens proteome database (2018). For the phosphoproteomics data, the searched parameters were set as follows: ^16^O was set as a light label and ^18^O was set as a heavy label. Trypsin/P was chosen as the protease and up to two missed cleavages were allowed. The variable modifications included acetyl (protein N-term), oxidation (M), and phospho (STY). Carbamidomethyl (C) was used as a fixed modification. The protein false discovery ratio (FDR) was set to 0.01. Using iGPS 1.0 software, a kinase substrate phosphorylation network (KSPN) was constructed by submitting differentially expressed phosphopeptides.

### 4.11. Migration and Invasion Assays

For the migration and invasion assays, 8 μm pore size cell culture inserts (#353097, BD Falcon, Franklin Lakes, NJ, USA) and Matrigel-coated inserts (#354480, Corning, NY, USA) were used. Cells were seeded in inserts, and 10% FBS containing medium was added to the lower chamber. After 24 h of incubation, the chambers were stained with a crystal violet solution. The stained cells were dissolved in DMSO and the absorbance was measured using a microplate spectrophotometer (wavelength 590 nm, BioTek Instruments, Winooski, VT, USA).

### 4.12. Quantitative Polymerase Chain Reaction (qPCR) Analysis

Total RNA extraction was performed using Maxwell^®^ RSC (Promega, Fitchburg, MA, USA), and GoScript™ Reverse Transcriptase (Promega, Fitchburg, MA, USA) was used to obtain cDNA according to the manufacturer’s protocol. Sequences of qPCR primers used were PDK4 Fw: 5′-TCT GAG GCT GAT GAC TGG TG-3′; Rv: 5′-CAG GAA GCA GCA CTG GTG TA-3′ and GAPDH FW: 5′-GTC TCC TCT GAC TTC AAC AGC G-3′; Rv: 5′- ACC ACC CTG TTG CTG TAG CCA A-3′. qPCR was performed using an ABI StepOnePlus thermocycler (Applied Biosystems, Foster City, CA, USA) with Luna Universal qPCR Master Mix (SYBR Green) purchased from NEB (Ipswich, MA, USA).

### 4.13. Experimental Animals

All animal study protocols were approved by the Institutional Animal Ethics Committee of Yeungnam University College of Medicine (YUMC-2020-026). All animals were kept in a controlled, specific pathogen-free environment under a 12 h light/dark cycle at a temperature of 25 ± 0.2 °C and humidity of 45 ± 2%. All mice were provided food and water ad libitum.

### 4.14. Immunohistochemistry (IHC) Staining

For IHC analysis, the deparaffinized slides were processed for antigen retrieval and blocking. Anti-PDK4, anti-phosphor-SRC, anti-phosphor-ERK1/2, and anti-phospho-JNK antibodies (1:100 dilution) were incubated overnight at 4 °C and then with the secondary antibody (Alexa Fluor 594, Thermo Fisher Scientific). Images were captured using a fluorescence microscope (Nikon Eclipse-80i, Tokyo, Japan).

### 4.15. Patient Samples

The patient samples were collected with informed consent and provided by the National Biobank of Korea-Kyungpook National University Hospital, a member of the Korea Biobank Network-KNUH, under Institutional Review Board-approved protocols (approval number: KNUCH 2022-01-006). The collected biospecimens were stored at −80 °C before use. PDK4 expression was analyzed by qPCR in 7 normal samples and 12 samples each from the Ta, T1, T2, and T3 groups.

### 4.16. Statistical Analysis

Values are expressed as mean ± SD. Analysis of variance and post hoc tests were used to assess the biological activity data, with a statistical significance level of *p* < 0.05. The differences between the treatment and control groups were compared using an unpaired Student’s *t*-test.

## Figures and Tables

**Figure 1 ijms-23-13240-f001:**
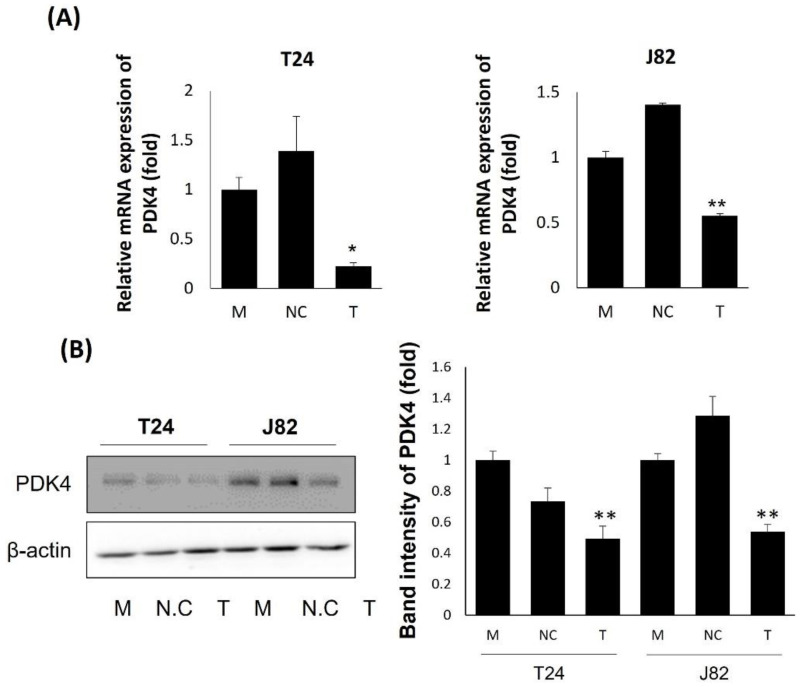
Expression of PDK4 in siRNA-treated bladder cancer cell lines. (**A**) Fold change of PDK4 mRNA expression in T24 and J82 cells. (**B**) Protein expression of PDK4 in T24 and J82 cells. All data represent means ± SD of three independent experiments (* *p* < 0.05, ** *p* < 0.01 NC vs. T). M: mock, N: negative control, T: PDK4 siRNA treated.

**Figure 2 ijms-23-13240-f002:**
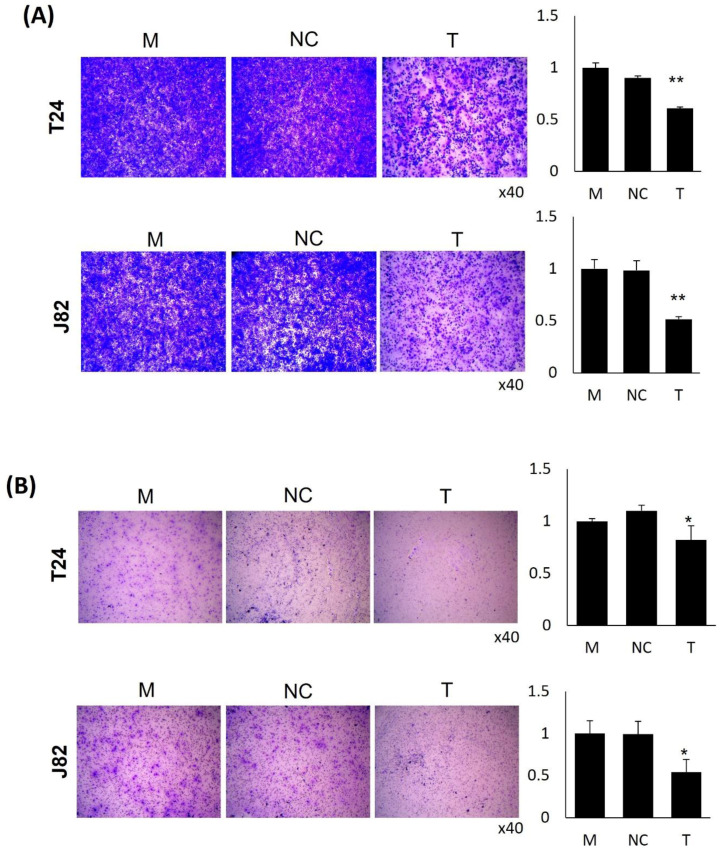
Migration and invasion assay of PDK4 knockdown bladder cancer cells. (**A**) Migration assay. (**B**) Invasion assay. All data represent means ± SD of three independent experiments (* *p* < 0.05, ** *p* < 0.01 NC vs. T). M: mock, N: negative control, T: PDK4 siRNA treated.

**Figure 3 ijms-23-13240-f003:**
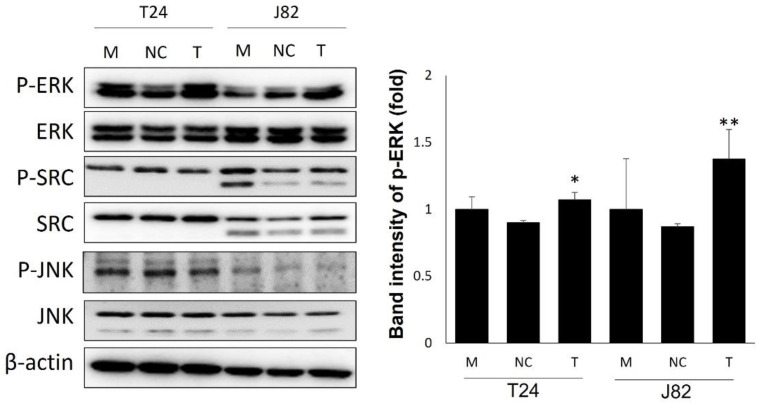
PDK4-related protein expression in PDK4 knockdown bladder cancer cells. β-actin is used as an internal control. M: mock, N: negative control, T: PDK4 siRNA treated. All data represent means ± SD of three independent experiments (* *p* < 0.05, ** *p* < 0.01 NC vs. T). M: mock, N: negative control, T: PDK4 siRNA treated.

**Figure 4 ijms-23-13240-f004:**
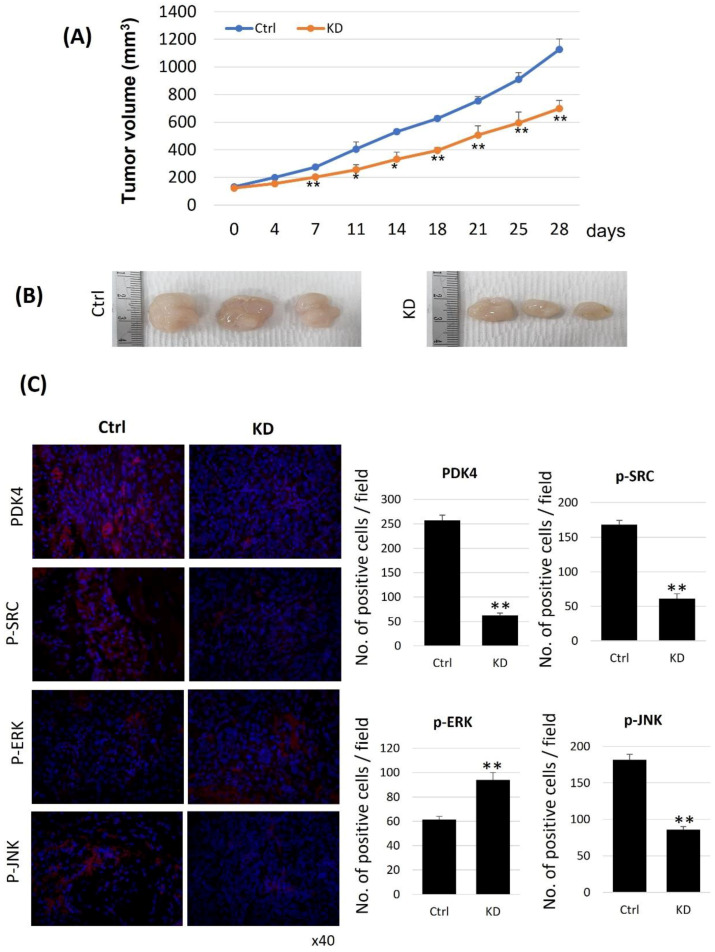
Xenograft model of PDK4 knockdown J82 cells. (**A**) Tumor growth. (* *p* < 0.05, ** *p* < 0.01 Day 0 vs. Each day). Ctrl: wild type, KD: knockdown. (**B**) Gross appearance. (**C**) Representative image of immunohisto-chemistry. (* *p* < 0.05, ** *p* < 0.01 Ctrl vs. KD).

**Figure 5 ijms-23-13240-f005:**
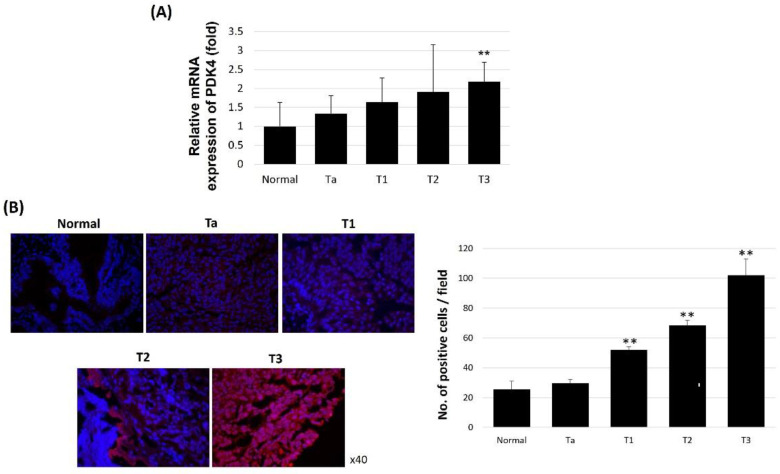
PDK4 expression in human bladder cancer specimen. (**A**) Relative PDK4 mRNA expression in normal and bladder cancer specimen. (**B**) Representative image of immunohisto-chemistry of PDK4 protein. All data represent means ± SD of three independent experiments (** *p* < 0.01 normal vs. T stages).

**Figure 6 ijms-23-13240-f006:**
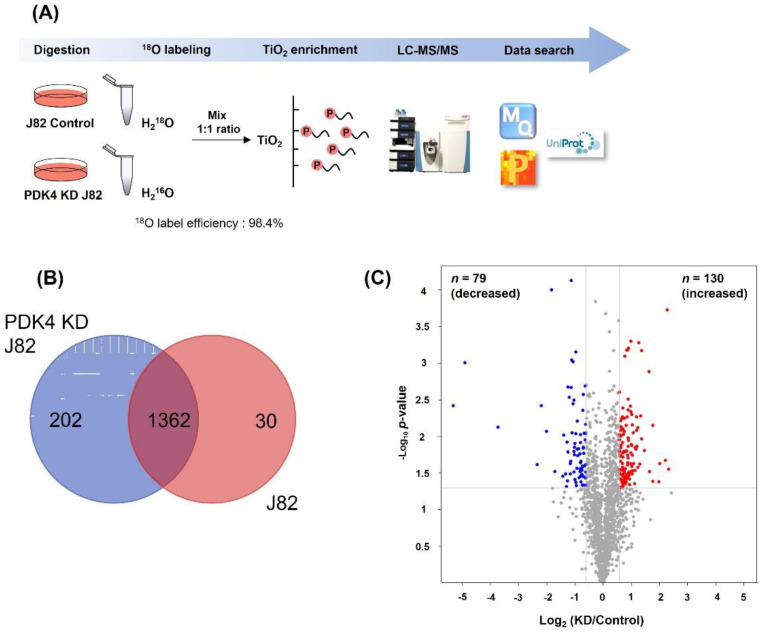
MS-based quantitative proteomic profiling of phosphorylation in bladder cancer cell lines. (**A**) Experimental design of proteomic analysis. J82 and J82 KD cells were lysed, subjected to in-solution digestion and ^18^O labeling, and then mixed at equal protein amounts. Phosphorylated peptides were enriched by TiO_2_. Eluted peptides were analysis using LC-MS/MS. Mass spectrum data were searched in the MaxQuant (version 1.5) database. (**B**) Venn diagrams showing overlap between J82 knock down and J82 control. (**C**) Volcano plot for 209 differentially phosphorylated protein (DRPs) between PDK4 knockdown and controls.

## Data Availability

Data available in a publicly accessible repository.

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
