# Peer review of "Anti-Metastatic Effect of Pyruvate Dehydrogenase Kinase 4 Inhibition in Bladder Cancer via the ERK, SRC, and JNK Pathways"

_ijms, 2022, doi:10.3390/ijms232113240_

Round 1

Reviewer 1 Report

Questions/Comments:

In this work, the authors evaluated the role of PDK4 in bladder cancer and its related protein changes. The data looks interesting and I have some questions listed here to be addressed.

1.     Why was there such a different response between the J82 cell line and the T24 cell line regarding expression of P-SRC and P-JNK? From the figure above, it is clear that there is a reduction of migration/invasion in the PDK4 inhibited group, it may not be these specific proteins that drive this change.

2.     PDK4 has some evidence showing it could be a biomarker for bladder cancer but many of the other protein changes from the paper are not uniform among the 2 different bladder cancer cell lines. The authors should consider including more replicates to have a deeper understanding of what markers exist in most types of bladder cancer.

3.     I would suggest performing proteomic analysis of the two cancer cell lines (J82 and T24) with the PDK4 inhibition as you could identify various upregulated/downregulated groups of proteins across both lines. Doing IHC staining of SRC, ERK, and JNC is a good start but investigating more proteins could reveal even stronger biomarkers.

4.     The authors showed that knocking down PDK4 in bladder cancer cell lines reduces tumor pathology. Did the experimenters identify any good targeting molecules for imaging/inhibition of PDK4?

5.     Data in Fig1B, Fig3, Fig4C, and Fig5B need quantitive analysis, at least 3 replicates.

Author Response

  1. Why was there such a different response between the J82 cell line and the T24 cell line regarding expression of P-SRC and P-JNK? From the figure above, it is clear that there is a reduction of migration/invasion in the PDK4 inhibited group, it may not be these specific proteins that drive this change.

à Thank you for your comment. Cell lines have their own characteristics like there are different type of tumor cell lines, thus, J82 and T24 show different response. Even if they are same epithelial cells for urinary bladder cancer, their origins such as gender and age is different, therefore, each cell line shows different response in migration and invasion assay. That is the reason why we use several cell lines in experiments. And this reflects the heterogenic characteristic of tumors.

  1. PDK4 has some evidence showing it could be a biomarker for bladder cancer but many of the other protein changes from the paper are not uniform among the 2 different bladder cancer cell lines. The authors should consider including more replicates to have a deeper understanding of what markers exist in most types of bladder cancer.

 à As we mentioned above, we thought that we have to consider heterogeneity of cancer. According to ATCC, J82 and T24 cell lines have difference in age, gender, transfection efficacy, tumor expressed specific antigen and so on. That is the reason why 2 different bladder cancer cell lines show variant protein changes.

  1. I would suggest performing proteomic analysis of the two cancer cell lines (J82 and T24) with the PDK4 inhibition as you could identify various upregulated/downregulated groups of proteins across both lines. Doing IHC staining of SRC, ERK, and JNC is a good start but investigating more proteins could reveal even stronger biomarkers.

à Thank you for your valuable comment. According to Fig. S1, T24 showed no specific change in immunoblotting analysis by using pan-phospho-S/T/Y antibody. And in vivo assay, T24 did not form tumor in nude mice. These are two reasons for performing the study with J82 cell line only.

  1. The authors showed that knocking down PDK4 in bladder cancer cell lines reduces tumor pathology. Did the experimenters identify any good targeting molecules for imaging/inhibition of PDK4?

à We revealed the relationship between PDK4 and Src through proteomic analysis other biomolecular analysis methods. For further studies, we will explore deeper experiment for more novel molecular image study by using PET-CT and other methods.

  1. Data in Fig1B, Fig3, Fig4C, and Fig5B need quantitive analysis, at least 3 replicates.

  à We newly added graphs in the mentioned Figures, as you pointed out.

Reviewer 2 Report

The author showed the clinical significance of PDK4 in bladder cancer. There are several concerns. Please check the following points.

1.     In introduction, the author addressed that the effect of PDK4 on bladder cancer has not been studied. However, as far as I searched, the role of PDK4 in bladder cancer has already analyzed (PMID: 29907593).

2.     In figure 1B, the band of PDK4 is unclear.

3.     In figure 3, it seems that the targeted molecules (p-SRC, p-JNK) were not different between negative control and siRNA for PDK4. Why did the author conclude that lower concentrations of p-SRC and p-JNK was observed in the J82 cell line only?

4.     In figure 4A, knockdown of PDK4 decreased the tumor growth. Please analyze the effect of PDK4 knockdown on cell growth in vitro.

5.     In figure 5, the author performed fluorescence immunostaining of PDK4. However, it is difficult to distinguish between normal and tumor in this way. Please analyze the standard immunohistochemistry.

Author Response

 [Reviewer 2]

  1. In introduction, the author addressed that the effect of PDK4 on bladder cancer has not been studied. However, as far as I searched, the role of PDK4 in bladder cancer has already analyzed (PMID: 29907593).

   à Thank you for your comment. The paper you mentioned (PMID: 29907593) revealed that PDK4 is related in cell cycle and apoptosis as well as chemoresistance. However, the present paper researched about metastatic effect of PDK4 in bladder cancer. The present study includes human sample analysis data, thus this paper is the unique study. In addition, we proceeded proteomic analysis to reveal the pathway and validated the marker.

  1. In figure 1B, the band of PDK4 is unclear.

  à Sorry for poor image. We replaced the band.

  1. In figure 3, it seems that the targeted molecules (p-SRC, p-JNK) were not different between negative control and siRNA for PDK4. Why did the author conclude that lower concentrations of p-SRC and p-JNK was observed in the J82 cell line only?

  à Thank you for your valuable advice. There are changes in band intensity. We inserted the quantitative analysis for the band intensity. In T24 cell lines, p-SRC and p-JNK were decreased, and in J82 cell lines, only p-JNK showed decreased expression. We corrected in 2.2 Mechanism of PDK4 in bladder cancer cell lines section.

  1. In figure 4A, knockdown of PDK4 decreased the tumor growth. Please analyze the effect of PDK4 knockdown on cell growth in vitro.

  à In in vitro proliferation assay, PDK4 inhibition was not effective in cell apoptosis or suppressed proliferation. However, we proceeded in vivo assay because living organism’s environment is different from cell line. Therefore, we tried the experiment, then, we found out that PDK4 inhibited cells shows lower proliferation rate in in vivo assay.

  1. In figure 5, the author performed fluorescence immunostaining of PDK4. However, it is difficult to distinguish between normal and tumor in this way. Please analyze the standard immunohistochemistry.

à We are very appreciated with your pointing out. In the fluorescence IHC results, we can observe normal epithelium and in tumor samples, we can observe that messed up histological findings like distracted nuclear arrangement.

Round 2

Reviewer 1 Report

The authors have addressed my questions.

Author Response

The authors have addressed my questions.

--> We are highly thankful to your review.

Reviewer 2 Report

The author did not appropriately respond to my question.

1.     According to figure 2, knockdown of PDK4 decreased migration and invasion, indicating PDK4 promotes migration and invasion activity. However, in figure 3, knockdown of PDK4 increased p-ERK. Increased p-ERK promoted oncogenic activity. There is a discrepancy.

2.     In figure 4, the expression of p-SRC and p-JNK in KD decreased compared to those in control. This finding was not consistent with the in vitro study.

Author Response

  1. According to figure 2, knockdown of PDK4 decreased migration and invasion, indicating PDK4 promotes migration and invasion activity. However, in figure 3, knockdown of PDK4 increased p-ERK. Increased p-ERK promoted oncogenic activity. There is a discrepancy.

--> ERK activation is still on going question in tumorigenesis.  In normal cell condition, ERK is responsible for cell proliferation and homeostasis, however, in tumor condition, the function of ERK is easily broken down and messed up. According to review papers about ERK in cancer (PMID 24408923, 32724338), ERK is reported to be dual functional protein. Therefore, in case of PDK4 related ERK, the authors concluded as increased p-ERK acted as tumor suppressor protein.

  1. In figure 4, the expression of p-SRC and p-JNK in KD decreased compared to those in control. This finding was not consistent with the in vitro study.

--> Both in vitro (Figure 3) and in vivo (Figure 4) show same tendency of p-SRC and p-JNK. p-SRC and p-JNK were decreased in siRNA treated group and PDK4 knockdown cell injected mice group. We hope your kind re-check of the figures.

Round 3

Reviewer 2 Report

The manuscript is well revised.